# Employment among Childhood Cancer Survivors: A Systematic Review and Meta-Analysis

**DOI:** 10.3390/cancers14194586

**Published:** 2022-09-22

**Authors:** Alessandro Godono, Francesco Felicetti, Alessio Conti, Marco Clari, Margherita Dionisi-Vici, Filippo Gatti, Catalina Ciocan, Tommaso Pinto, Emanuela Arvat, Enrico Brignardello, Franca Fagioli, Enrico Pira

**Affiliations:** 1Department of Public Health and Pediatrics, University of Torino, 10126 Turin, Italy; 2Transition Unit for Childhood Cancer Survivors, Azienda Ospedaliera Universitaria Città della Salute e della Scienza di Torino, University of Torino, 10126 Torino, Italy; 3Oncological Endocrinology Unit, Department of Medical Sciences, University of Turin, Città della Salute e della Scienza Hospital, 10126 Turin, Italy; 4Stem Cell Transplantation and Cellular Therapy Laboratory, Paediatric Onco-Haematology Division, Regina Margherita Children’s Hospital, City of Health and Science of Turin, 10126 Torino, Italy

**Keywords:** cancer survivors, neoplasm, paediatric, childhood, transition, employment, socio-economic status

## Abstract

**Simple Summary:**

Despite the improvements in the survival rates and functional outcomes of childhood cancer survivors (CCS), most of them experience late effects with possible consequences to their occupational status. To date, a reliable estimate of the prevalence of employment among this population is still missing. This study aimed to assess, for the first time, the prevalence of employment among CCS and to examine the associations of socio-demographic and clinical factors with employment. Almost 100 cohorts worldwide have been included in this review, highlighting that two-thirds of childhood cancer survivors were employed. Different employment rates depending on socio-demographic and clinical factors were identified. The findings from this study could facilitate the design of targeted interventions aimed at promoting employment among CCS.

**Abstract:**

To date, there are heterogeneous studies related to childhood cancer survivors’ (CCS) employment rates. Given the importance of this topic, we aimed to perform a systematic review and meta-analysis to investigate the prevalence of employment among CCS and to examine its association with socio-demographic and clinical factors. We followed the PRISMA guidelines to search for pertinent articles in relevant electronic databases. Eighty-nine articles comprising 93 cohorts were included. The overall prevalence of employment was 66% (CI: 95% 0.63–0.69). Subgroup meta-analyses showed that lower rates were found for central nervous system tumor survivors (51%, CI: 95% 0.43–0.59), and for CCS treated with cranial-radiotherapy (53%, CI: 95% 0.42–0.64) or haematopoietic stem-cell transplantation (56%, CI: 95% 0.46–0.65). The studies conducted in Asia highlighted employment rates of 47% (CI: 95%, 0.34–0.60). Univariate meta-regressions identified the following socio-demographic factors associated with higher rates of employment: a female gender (*p* = 0.046), a higher mean age at the time of investigation (*p* = 0.00), a longer time since diagnosis (*p* = 0.00), a higher educational level (*p* = 0.03), and a married status (*p* = 0.00). In conclusion, this systematic review and meta-analysis provides evidence that two-thirds of CCS are employed worldwide. Identifying vulnerable groups of CCS may allow for the design of multidisciplinary support strategies and interventions to promote employment in this population.

## 1. Introduction

The survival rate for childhood cancer has considerably improved during the last few decades, and is now approaching 80% [1] due to substantial advances in diagnostics and treatment strategies [2]. Despite these improvements, two out of three Childhood Cancer Survivors (CCS) will experience at least one late effect (LE), while approximately 40% of them will experience severely disabling life-threatening or fatal clinical conditions over 30 years from diagnosis [3,4]. Nevertheless, CCS are a population at high risk for disrupted psychosocial development secondary to their primary disease, treatment, and physical LEs [5]. Previous studies highlighted that a childhood cancer diagnosis might negatively influence school performance, educational achievements, social life, and marital status [6]. Moreover, a recent study showed that one in six CCS are unemployed and that they are 1.5 times more likely to be unemployed than healthy controls [7]. Less is known about the CCS’s actual employment rate, which is currently considered a more reliable measure to assess trends in the occupational market. According to the definition of the International Labour Organisation (ILO), the term “employed” comprises all persons above a specified age (usually 15 years old) who, during a short reference period, were in one of the following categories: paid employment (*at work* or *with a job but not at work*) or self-employment generating an economic profit (*at work* or *with an enterprise but not at work*) [8]. The current scientific literature about CCS’s employment is heterogeneous along with the broad spectrum of childhood cancers in addition to country-specific educational systems.

Given the paucity of secondary literature on CCS’s employment status, yet, at the same time, its importance in terms of social impact, the purpose of this systematic review and meta-analysis is to provide comprehensive data on the prevalence of employment among CCS and to examine the associations of socio-demographic and clinical factors with employment rates.

## 2. Materials and Methods

This systematic review and meta-analysis was performed and reported according to the Preferred Reporting Items for Systematic Reviews and Meta-Analyses (PRISMA) guidelines [9]. The review protocol was registered on the international prospective register of systematic reviews (PROSPERO; CRD42022344410). This article presents aggregate data from primary studies; thus, no ethical approval was requested.

We conducted a systematic search for pertinent articles in five relevant electronic databases: Medline PubMed, Web of Science, Embase, Cochrane, and PsycINFO from their inception to May 2022. No limits were applied for language, and results were limited to studies conducted on humans. Search strings included terms related to the occupational field such as “employment, unemployment, absenteeism, presenteeism, work productivity, work capacity, work engagement, work ability, work performance, workload, workplace, job satisfaction, sickness absence”, and they were combined with the population of interest with terms such as “neoplasm, cancer, carcinoma […]” and “survivors, childhood survivors, paediatric […]”. The search strategy was firstly launched on PubMed and then adapted for all databases. (Appendix A). An expert librarian was involved in the database searches to ensure methodological rigor. The reference lists of included articles were also manually screened to identify further relevant articles. The literature search was conducted independently by three investigators and each abstract was evaluated in duplicate by two investigators. Full reports of potentially relevant articles were evaluated independently by two investigators. Disagreements were resolved between investigators and with the help of a third reviewer through consensus.

To be included in this systematic review, studies were required to be primary investigations based on a sample of CCS. Studies were included if they met the following criteria:-Included patients with a previous diagnosis of childhood cancer;-Mean age of 18 years or higher at the time of investigation;-Mean age of 16 years or lower at the time of the diagnosis;-Presented data on the employment status of included patients.

A proven diagnosis of childhood cancer, total number of CCS, and employment status of CCS were considered mandatory variables for inclusion. Only articles published in peer-reviewed journals were considered. Experimental studies, other systematic reviews or meta-analyses, and conference proceedings, theses, and letters to the editor were excluded. Articles for which the full text was not available either online or following request to the journal in which they were published were excluded.

For each cohort, the number of employed CCS was extracted as the primary outcome measure. An employed participant was intended as someone with paid employment who, during the article reference period, worked for at least one hour during a given week or had a job from which being absent was conditional on the reason of absence (e.g., holidays, maternity leave, etc.) or duration [8]. In the case of multiple reports from the same cohort, the most complete results (i.e., those based on the largest number of cases) were used. The following study characteristics were also extracted if reported in the article: publication year, country, study design, cohort size, number of males, cancer type, treatment type, mean age at the time of investigations, mean age at diagnosis, duration of the follow-up, ethnic groups, marital status, education level, number of students, and presence of a control group. Data were extracted by three independent reviewers, and any disagreement was solved by a fourth reviewer.

The CCS diagnoses were sorted into diagnostic groups: multiple cancers, central nervous system (CNS) cancers, haematological cancers, bone cancers/sarcomas, and thyroid cancers. Cohorts have been categorised in a treatment regimen if more than 50% of the CCS were treated with specific therapy, subdivided into radiotherapy (RT), cranial RT, stem-cell/bone marrow transplantation, and surgery. When data on mean age were not directly reported, they were calculated through quantile estimation [10].

Methodological quality was assessed using the Checklist for Prevalence Studies by the Joanna Brigs Institute [11], a nine-question tool with four standard answer options divided into four main domains (population and setting, condition measurement, statistics, and other), which allowed for the execution of a series of subgroup meta-analyses to assess the difference in prevalence reported by studies with different quality. As the assessment tool did not provide cut-off values, the average and median scores (M = 6.16; median = 6) were calculated to define the poor, fair, and good quality of articles. The quality assessment (QA) score of the articles was rated as poor (score = 4), fair (score = 5, 6, and 7), and good (score = 8). QA was performed by three independent reviewers, and results were discussed with a fourth reviewer until reaching consensus. The criteria were tested on a set of 10 articles to ensure agreement between assessors. Since the articles often considered employment status as a socio-demographic variable, studies were not excluded based on the QA scores. 

### 2.1. Statistical Analysis

#### Overall Pooled Prevalence of Employment in CCS

Before conducting the overall pooled prevalence meta-analysis, the heterogeneity of prevalence estimates was assessed by calculating the I^2^ index and performing the Cochran Q test. An I^2^ > 50% and Cochran Q test *p*-values < 0.05 represented a high degree of significant heterogeneity. Due to the high heterogeneity that was both found and expected, we performed a random-effects meta-analysis of employment among CCS with 95% Confidence Intervals (CIs). As in highly heterogeneous meta-analyses, the random-effects model still has a high mean squared error; a meta-analysis using a quality effect estimator was also performed.

Sensitivity analyses included repetitions of the main meta-analysis; in each repetition, one article was removed to observe any individual effects. In addition, a subgroup meta-analysis by QA scores (poor, fair, and good) was performed to assess the variability between QA scores.

We assessed the presence of publication bias and small study effects by visual inspection of the funnel plots and applying the test proposed by Egger et al. [12].

### 2.2. Subgroup Meta-Analyses

We conducted subgroup meta-analyses to determine potential sources of heterogeneity. Four subgroup meta-analyses were performed to assess the prevalence of employed CCS according to different cancer diagnoses (grouped into five categories: multiple cancers, central nervous system, haematological cancers, bone cancers/sarcomas, and thyroid cancers treatment types), treatment types (multiple, mainly surgery, mainly RT, mainly cranial RT, and mainly stem-cell transplantation), and geographical areas (North America, Europe, and Asia). Data from at least three studies should be available to perform subgroup analyses.

### 2.3. Meta-Regressions

We performed a series of meta-regressions to examine the association between socio-demographic and clinical factors with respect to employment. The following parameters were investigated: mean age of the study group participants, mean age upon diagnosis, time since the diagnosis, percentages of males/females, percentage of participants that have graduated, and marital status. Firstly, we analysed the association of these variables in a univariate analysis. Variables statistically significantly associated with CCS’s employment were included in a multivariate analysis using a random-effects meta-regression model. Data on CCS characteristics from at least ten studies should be available to perform a univariate meta-regression and 20 for a multivariable meta-regression.

Data analyses were conducted using STATA SE/17 (StataCorp LLC, College Station, TX, USA).

## 3. Results

The database search yielded a total of 6525 articles. After the duplicates were removed (*n* = 1082), 5443 articles remained. After reviewing the articles by titles and abstracts, 261 articles were considered relevant for inclusion. The full texts of these articles were examined in detail and assessed against the inclusion and exclusion criteria. A manual search of the reference lists of the included articles did not reveal additional relevant studies.

Eighty-nine articles [13,14,15,16,17,18,19,20,21,22,23,24,25,26,27,28,29,30,31,32,33,34,35,36,37,38,39,40,41,42,43,44,45,46,47,48,49,50,51,52,53,54,55,56,57,58,59,60,61,62,63,64,65,66,67,68,69,70,71,72,73,74,75,76,77,78,79,80,81,82,83,84,85,86,87,88,89,90,91,92,93,94,95,96,97,98,99,100,101] met the inclusion criteria for the systematic review and meta-analysis. These articles reported on a total of 93 cohorts of CCS. The screening process is summarised in Figure 1.

Forty-three studies were conducted in North America, forty in Europe, and six in Asia. The articles were published between 1989–2022.

The total number of CCS was 123,734. The mean age at the time of investigations was reported in 81.7% of the cohorts (27.99 years; SD ± 5.38), and the mean age at diagnosis was 9.12 years (SD ± 2.58), reported in 66.7% of the cohorts. The mean time since the diagnosis was 18.34 years (SD ± 5.27), reported in 66.7% of the cohorts. The gender was specified in 91.4% of the cohorts (50.93% males; SD ± 10.67), while marital status was specified in 63.4% of the cohorts (34.64% married; SD ± 15.96). The percentage of students was reported in 34.4% of the cohorts (mean 24.8%; range 2.5–51.3).

A total of 52 out of the 93 cohort studies analysed multiple cancer types, while 41 focused on a specific origin, namely CNS (*n* = 18), the haematopoietic system (*n* = 18), bone/soft tissue (*n* = 4), and thyroid gland (*n* = 1). In 40 of the 93 cohorts, CCS were mainly treated (>50% of the total population) with RT (*n* = 12), cranial-RT (*n* = 12), surgery (*n* = 11), and stem-cell/bone marrow transplantation (*n* = 5). Fifty-five studies (60%) reported data for CCS diagnosed and treated before 1990, whereas twenty studies (22%) reported data for after 1990. Seventeen articles (18%) did not report this information. Forty-seven studies enrolled a control population: 21 among siblings, 24 among the general population, and 2 from both. Table 1 summarises the characteristics of the included studies.

The overall quality of the included studies was fair. Particularly, nine articles attained a high QA, whereas six were of poor quality, and the remaining seventy-four were of fair quality. The items that received a higher number of negative answers were related to the description of the setting and participants (question 4), the measurement of the condition in a standardised and reliable way for all participants (question 7), and the appropriateness of the applied statistical analysis (question 8). These were related to the scarce reporting and measurement of CCS’s employment rates, with their related clinical and socio-demographic characteristics, which were clearly stratified among those employed and unemployed. No QA questions were deemed unapplicable to the included articles. The complete quality assessment is reported in the Appendix A.

### 3.1. Meta-Analyses

The overall prevalence of employment among CCS was 66% (95% CI, 0.63–0.69). The pooled prevalence of employment, stratified by cancer type, is shown in Figure 2. The lowest prevalence was found for CNS tumours (51%; 95% CI, 0.43–0.59), followed by haematologic malignancies (65%; 95% CI, 0.53–0.76), multiple cancers (68%; 95% CI, 0.65–0.72), bone cancer/sarcoma (81%; 95% CI, 0.76–0.86), and thyroid cancer (91%; 95% CI, 0.82–0.97).

Subgroup meta-analyses by type of treatment showed the lowest prevalence of employment in the cohorts of CCS mainly treated with cranial RT (53%; 95% CI, 0.42–0.64) or haematopoietic stem-cell/bone marrow transplantation (56%; 95% CI, 0.46–0.65), whereas they found the highest in the cohorts of CCS mainly treated with surgery (77%; 95% CI, 0.72–0.82). CCS diagnosed and treated before 1990 had higher employment rates (72%; 95% CI, 0.68–0.75) than those diagnosed and treated after 1990 (50%; 95% CI, 0.41–0.59). Finally, with regard to geographical differences, studies conducted in North America showed a prevalence of employment of 73% (95% CI, 0.70–075), in Europe of 60% (95% CI, 0.53–0.67), and in Asia of 47% (95% CI, 0.34–0.60).

There was evidence of significant heterogeneity (I^2^ > 50%; *p* = 0.00) in all the meta-analyses performed.

The funnel plot for the overall meta-analysis was scattered and asymmetrical, representing the possible presence of reporting bias. Similarly, the results of Egger’s tests were statistically significant for the presence of a small study effect.

### 3.2. Factors Associated with Employment

A univariate meta-regression (Table 2) identified the socio-demographic factors associated with a higher prevalence of employment: female gender (*p* = 0.046), higher mean age at the time of investigations (*p* = 0.00), longer time since diagnosis (*p* = 0.00), higher educational level (*p* = 0.03), and being married (*p* = 0.00). Moreover, the crude univariate meta-regression for Europe (*p* = 0.00) and Asia (*p* = 0.002) highlighted a lower employment prevalence than North America.

The clinical factors associated with lower employment prevalence were as follows: a diagnosis of a CNS tumor (*p* = 0.00), cranial RT (*p* = 0.021), CNS tumours treated with cranial RT (*p* = 0.00), haematologic malignancy treated with haematopoietic stem cell/bone marrow transplantation (*p* = 0.011), and a diagnosis and treatment after 1990 (*p* = 0.00).

In the multivariate analysis (Table 2), the prevalence of employment after adjustment for the age, gender, and geographical area was significantly lower for CNS tumours (*p* = 0.006) compared to the multiple cancer studies. The employment rates were lower for Asian studies compared with those conducted in Europe and North America after adjusting for the mean age, gender, and cancer type.

### 3.3. Sensitivity Analyses

The omission of any single article from the main meta-analysis did not influence the pooled prevalence of employment, with a maximum variation in the outcome of 1% (*p* < 0.01). The subgroup analyses by quality score revealed that the articles with a high methodological quality reported a prevalence of employment of 73% (95% CI, 0.64–0.81), while studies with a low-quality score had a 48% (95% CI, 0.35–0.62) prevalence. The majority of the articles had a fair quality score with a prevalence of 66% (95% CI, 0.63–0.69), consistent with the overall pooled prevalence found.

## 4. Discussion

This systematic review and meta-analysis determined that two-thirds of CCS are employed in their adulthood. However, the overall prevalence found is highly dependent on the cancer diagnosis and treatment. While almost all the survivors of bone cancer and sarcoma are employed, only half of those CCS who suffered from CNS tumours have a gainful occupation. Cranial RT and stem cell transplantation have worse outcomes, with approximately half of CCS treated with such therapies being employed. On the other hand, CCS subjected to surgery showed considerably higher employment rates, as well as those diagnosed and treated before 1990. A greater age, a longer time since diagnosis, a higher degree of education, a female gender, and being married were factors that were significantly associated with a higher rate of employment. Asian and European studies reported a significantly lower employment prevalence when compared to those conducted in North America.

To the authors’ knowledge, this is the first study assessing the overall employment prevalence estimate among CCS. Previous reviews on this population have considered their occupational status from an unemployment perspective. In particular, Mader et al. estimated that CCS were 1.5 times more likely to be unemployed than the healthy controls [7]. This value is even more favorable than those found by De Boer et al. in 2006, who highlighted that CCS were twice as likely to be unemployed as controls [102]. The quite elevated employment prevalence highlighted in this study, especially for some CCS groups, may represent a direct effect of the improvement in the safety of anticancer treatments and of recently implemented welfare policies aimed at improving social outcomes among CCS [103,104]. Indeed, the obtained findings for CCS are consistent with those found in 2009 among adult survivors, considering their unemployment rate of 33.8% [105], and recently among adolescent and young adult survivors, reporting an 84.4% lifetime prevalence of employment [106]. Nevertheless, the comparisons between employed and unemployed prevalence rates should be considered carefully, as these concepts are not completely complementary.

Consistently with previous reviews [7,102], a low prevalence of employment was found among CNS tumour survivors, while haematological malignancies, bone cancers, and sarcomas had better occupational outcomes [102]. These findings are not unexpected, as survivors of childhood-onset CNS tumours are almost five times more likely to be unemployed than healthy controls [7]. In particular, unemployment rates in these subjects ranged from 25 to 50% [102], and similar adverse employment outcomes have been reported for adult CNS cancer survivors [105]. Indeed, these patients are prone to suffer neurocognitive disorders, memory, and mobility limitations [107], which represent essential functions for workers. Moreover, they have limited chances of obtaining managerial or professional employment and high incomes [6]. Given their poor employment outcomes, childhood survivors of CNS tumours should receive specific training considering their impairments and benefit from sheltered employment opportunities.

Cranial RT plays a critical role in treating CNS tumours [108]. In addition to the development of numerous LEs such as fatigue and neurocognitive deficits, this therapeutic approach has already been identified as a predictor of unemployment among CCS [7]. Moreover, it has been shown that children who gained employment despite being treated with cranial RT reported reduced incomes [6]. Similarly, haematopoietic stem cell transplantation might result in more LEs and reduced growth among CCS, leading to higher unemployment rates among these individuals [109]. Furthermore, our findings showed no substantial differences in the prevalence of employment in CCS who underwent RT or chemotherapy, even if Ketterl et al. [106] reported increased levels of mental impairment in the work tasks among these groups.

The higher prevalence of employment among CCSs in North America compared with Europe and Asia would seem inconsistent with the data reported elsewhere in the literature. In particular, a recent review showed twofold odd unemployment rates for CCS from North America compared with healthy controls, which were higher than those observed in Europe [7]. The same trend was observed in adult cancer survivors [105]. However, as found in our study, the risk of unemployment in North American studies on adult survivors was no different from European studies after an adjustment for sociodemographic and clinical variables. The possible explanations regarding the higher values found in North America are related to the elevated occurrence of temporary or part-time jobs and the health insurance system [102]. Health insurance in North America is primarily provided and covered financially by employers [110] and could represent a crucial factor for CCS concerning their choice to work to ensure access to lifelong health care. On the other hand, employers in Europe ensure more flexible working conditions, helping survivors in maintaining their employment status [111,112]. Most European studies included were from Scandinavia, where social welfare systems are well-developed [6]. However, despite the positive aspects of social welfare systems, they can disincentivise CCS from seeking work, making unemployment financially attractive [113]. The significantly lower employment prevalence found in Asia compared with North America and Europe may reside in the Eastern culture of the preservation of vulnerable individuals or be related to the existence in these contexts of specific legislation to protect cancer survivors from employment discrimination [114].

Age is a significant determinant of employment for the general population, with favourable outcomes reported in people aged 25–54 [115]. In CCS, a younger age at diagnosis and at the time of investigation has already been associated with unemployment [102]. Similarly, long-term CSS have a more successful recovery from LEs and showed no differences in absenteeism when compared to the healthy controls [83]. In contrast, a greater degree of presenteeism was reported among employed CCS with a longer time since diagnosis [83]. Therefore, it is not surprising that, in our study, a greater age and longer time since diagnosis have been associated with better employment outcomes. This finding could also explain the higher prevalence of employment found in CCS diagnosed and treated before 1990. Despite diagnostic and therapeutic advances in the last three decades, recently diagnosed CCS may still be young and already part of the working population. Regarding the high level of educational attainment by CCS, this could be hindered by treatments and their LEs, besides the delayed school progression [116]. This is of concern, as one’s educational level positively influences their possibility of employment and a higher income [117]. Compared with the general population, CCS showed lower academic success and marriage rates [7,118]. Moreover, among this population, a high frequency of divorce or separation has been reported, presumably influenced by financial stressors due to limited employment and a reduced income [119]. Thus, marital status or togetherness could represent protective factors for maintaining employment, since even in the case of reduced earnings partners could provide a motivation to continue working actively. Although several studies reported that female childhood survivors experience more health-related challenges towards employment [7], societal patterns suggest that they are more likely to achieve higher grades and educational success than males [6]. This could place them in a better position to acquire better employment and increased incomes, with male CCS more frequently employed at the manufacturing level and at a higher risk in terms of socioeconomic outcomes [6]. Indeed, CCS often require a physical component to perform their jobs, which is inevitably affected by the LEs of the treatment they were subjected to [106]. Nevertheless, although the obtained findings have shown better employment outcomes among females, the complexity of the labor market and the possible discrimination that CCS may face at present necessitate strategies that are above simple gender considerations.

Lastly, we found significant differences in the employment rates by conducting subgroup meta-analyses stratified by QA. The lower prevalence rates shown in the studies with poorer quality may be related to a lower sensitivity of the instruments used to assess employment among CCS. Moreover, among such studies, employment was not collected as a primary outcome. On the other hand, the studies with higher quality included large cohorts of participants, often stratifying the results between those CCS who were employed and those unemployed, thus generating a more reliable estimate of the prevalence.

### 4.1. Implications for Practice and Research

Despite the widespread emphasis on the need to develop interventions to promote social outcomes in CCS, to date, there is little evidence to guide healthcare providers in supporting these patients and their families. Multidisciplinary interventions involving physical, psycho-educational, and vocational components have shown efficacy in promoting adult cancer survivors’ return to work [105]. Similarly, interventions directed at CCS should be conducted with a multicomponent approach, focusing on clinical aspects and educational and social dimensions that may enable CCS to enter the labor market more confidently. These interventions should include a gradual approach to employment, the presence of referral figures over a lifetime to ensure work maintenance, and the opportunity to benefit from sheltered training periods in case of late effects’ occurrence or recurrence. Beyond the impact on employment, cancer has long-term effects on workability and work capacity, resulting in a potential reduction in income and the loss of life satisfaction for large groups of survivors [83,105]. There is a need to develop and prove the effectiveness of clinical and social support services focused on rehabilitation and workplace accommodation. The use of technology may represent a way to promote access to work and improve communication, by keeping young survivors and their families connected to healthcare and social providers [104]. Long-term follow-ups for CCS may lead to better health and educational outcomes [6]. Considering the results of our study, some CCS groups might benefit from a more focused follow-up and the implementation of appropriate support strategies [120]. These could increase the possibilities for CCS to be aware of the adverse socioeconomic situations that lead to unhealthy lifestyles and, consequently, more comorbidities [121]. From a social point of view, employment stability in CCS should be enhanced, as it is crucial for economic reasons and to prevent inequalities [122]. Better estimates of employment from high-quality studies involving large samples of CCS with matched control groups are needed to enable the reliable identification of vulnerable subgroups and the design of tailored interventions to promote employment among this population. Finally, future studies should apply reliable methodological standards to measure employment rates among CCS, ensuring the better comparability and statistical validity of the associated research. Particularly, they should report more accurately on participants’ clinical and socio-demographic characteristics, which are clearly differentiated among those employed and unemployed. 

### 4.2. Strengths and limitations

This systematic review and meta-analysis has some limitations. The search of five databases could have excluded some relevant studies. Furthermore, all the studies included were conducted in high-income countries, thereby limiting the generalizability of their results. Nevertheless, the application of a systematic approach and the involvement of three independent researchers and an expert librarian in all the phases (the search, screening, and extraction) contributed to limiting the biases related to the selection of articles.

The quality of the included studies ranged widely, and strictly depended on their design and objectives. It is possible that, as reported by Mader and colleagues [7], there may be a tendency in uncontrolled studies to overestimate the phenomenon of interest. In this regard, all the studies with low quality have a small sample size (less than 100 CCS), and half of them were published before 2000, so it is arguable that the QA scores could be directly affected by an earlier period of a given study’s execution, beyond a limited recruited sample. Moreover, in some studies, employment was only a secondary outcome that was always reported in an unreliable form, whereas other studies examined CCS longitudinally with matched control groups, leading to heterogeneity of the quality of reported data. To address this limitation, we extracted all the information about employment status, excluding all students, homemakers, and retired CCS from our analyses, attempting to include only those effectively employed. As this study included a very large number of articles, we believe that a large proportion of the studies on employment have been selected by our inclusion criteria.

The issues about the estimates of the phenomenon of interest and the high heterogeneity found in the meta-analyses were reduced by stabilizing the prevalence variance using a double-arcsine transformation [123]. Moreover, all the performed meta-analyses used random-effect models. Using a quality effects estimator could have maintained a lower estimated variance while maintaining the correct confidence interval probability, regardless of the level of heterogeneity. We addressed this limitation by stratifying the studies by quality level, which allowed us to assess the differences that occurred in the prevalence rates obtained from the quality assessment scores. In this regard, higher quality studies reported a higher prevalence and might have used measures targeted to assess the employment rate. Conversely, lower quality studies could have assessed employment as a secondary outcome, limiting the reliability of the detection of employment prevalence. Lastly, the presence of a small study effect due to the heterogeneous number of CCS included in the cohorts could be partly explained by the variations in the cancers’ occurrence and contextual differences among different geographical areas.

Despite the above limitations, this systematic review focused specifically on employment among CCS, stratifying its findings and providing evidence for the association of employment with clinical and sociodemographic variables by using robust meta-analysis methods.

## 5. Conclusions

This is the first systematic review and meta-analysis assessing the prevalence of employment among CCS, which highlights difference rates depending on socio-demographic and clinical factors. A greater age at investigation resulted in a higher rate of employment among CCS, while CNS tumors were associated with worse occupational outcomes. The studies from Asia reported significantly lower employment rates. Identifying susceptible groups of CCS may facilitate the design of multidisciplinary support strategies and multicomponent interventions focused on clinical and social aspects to promote employment in this population. Future research should employ longitudinal, controlled, matched designs focusing on specific cancer diagnoses and treatments to ensure more reliable employment estimates among CCS.

## Figures and Tables

**Figure 1 cancers-14-04586-f001:**
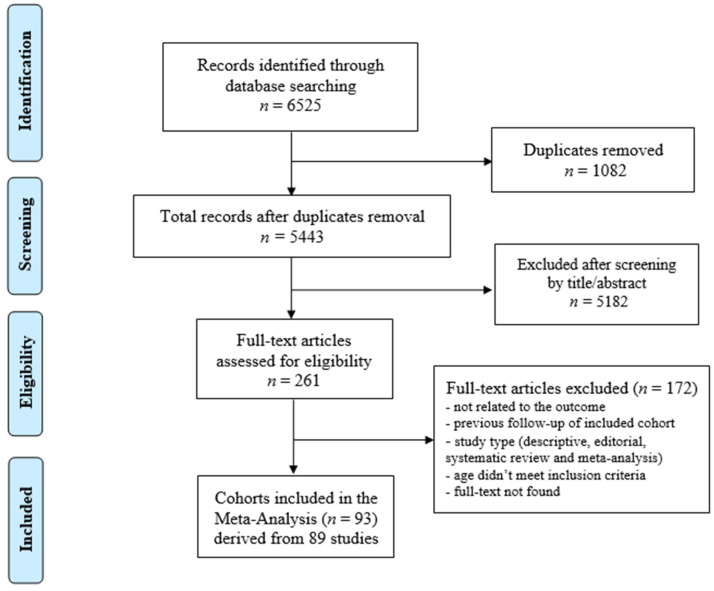
Preferred reporting items for systematic reviews and meta-analyses.

**Figure 2 cancers-14-04586-f002:**
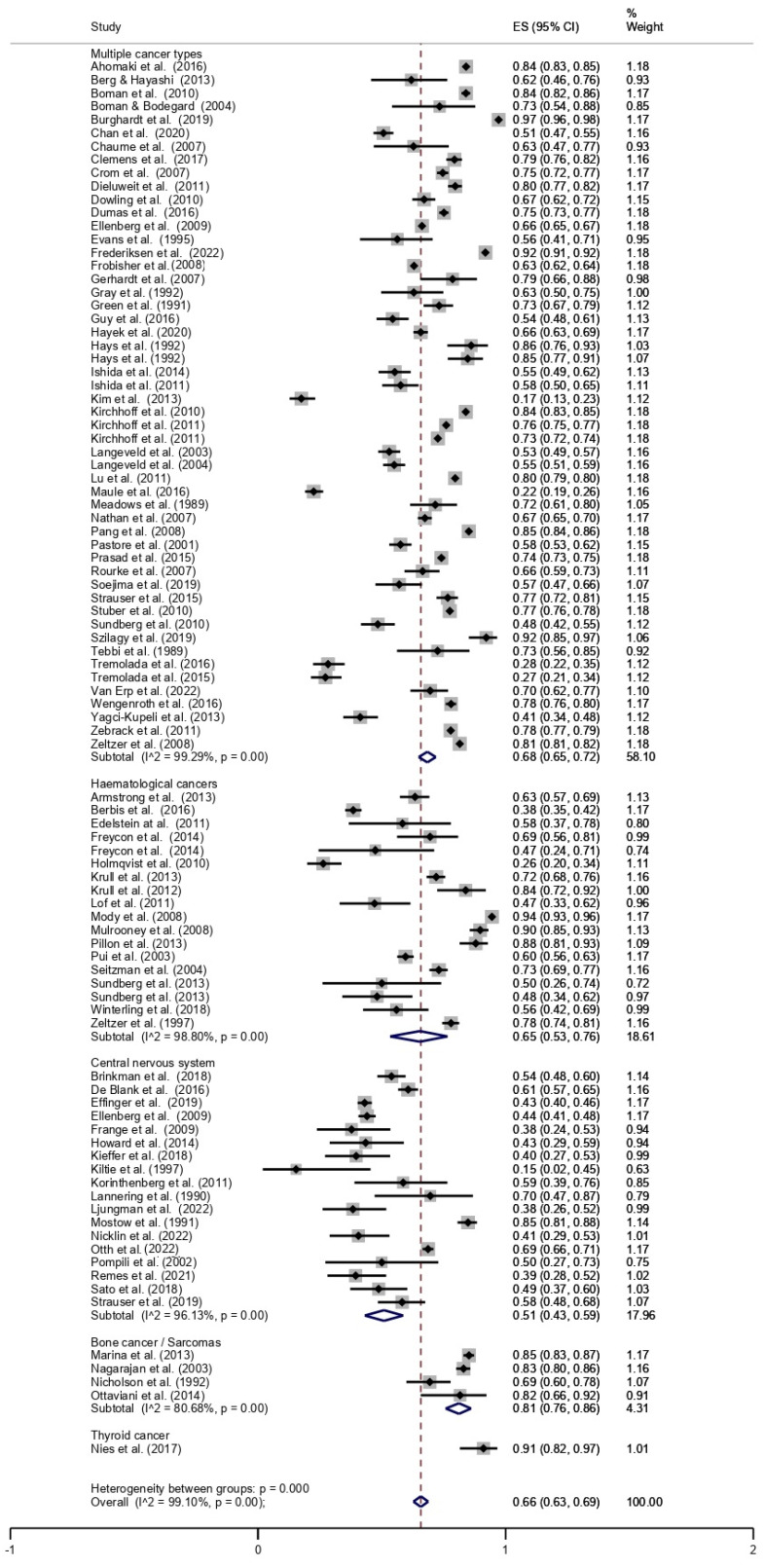
Forest plot stratified by cancer type [13,14,15,16,17,18,19,20,21,22,23,24,25,26,27,28,29,30,31,32,33,34,35,36,37,38,39,40,41,42,43,44,45,46,47,48,49,50,51,52,53,54,55,56,57,58,59,60,61,62,63,64,65,66,67,68,69,70,71,72,73,74,75,76,77,78,79,80,81,82,83,84,85,86,87,88,89,90,91,92,93,94,95,96,97,98,99,100,101].

**Table 1 cancers-14-04586-t001:** Characteristics of included studies.

Author	Year	Country	CCS (*n*)	Male (*n*)	Employed (*n*)	Cancer Type	Treatment Type *	Mean Age (At Investigation)	Mean Age (Diagnosis)	Married (*n*)
Ahomaki et al. [13]	2016	Finland	3243	1680	2725	Multiple	Multiple	29.01	-	-
Armstrong et al. [14]	2013	US	265	128	168	Haemato	Cranial RT	37.1	-	-
Berbis et al. [25]	2016	France	845	431	325	Heaemato	Multiple	22.3	7.9	-
Berg & Hayashi [36]	2013	US	42	14	26	Multiple	Multiple	20.5	9.8	-
Boman et al. [47]	2010	Sweden	1716	879	1441	Multiple	Multiple	31.6	-	-
Boman & Bodegard [58]	2004	Sweden	30	15	22	Multiple	Multiple	21.6	8.29	5
Brinkman et al. [69]	2018	US	306	174	165	CNS	Cranial RT	26.3	8.7	87
Burghardt et al. [80]	2019	Germany	951	526	924	Multiple	Multiple	34.49	4.99	354
Chan et al. [91]	2020	China	614	360	311	Multiple	Multiple	21.9	-	40
Chaume et al. [101]	2007	France	43	27	27	Multiple	Surgery	23.3	8	-
Clemens et al. [15]	2017	The Netherlands	653	366	518	Multiple	Multiple	26.25	6.12	188
Crom et al. [16]	2007	US	1437	719	1072	Multiple	RT	31.3	6.79	651
De Blank et al. [17]	2016	US	587	285	356	CNS	Multiple	-	-	205
Dieluweit et al. [18]	2011	Germany	820	402	653	Multiple	Multiple	29.9	15.8	-
Dowling et al. [19]	2010	US	410	173	275	Multiple	Multiple	-	-	193
Dumas et al. [20]	2016	France	2066	1058	1551	Multiple	RT	36	6	-
Edelstein at al. [21]	2011	Canada	24	8	14	Heaemato	RT	22.9	7.2	6
Effinger et al. [22]	2019	US/Canada	1182	632	509	CNS	Cranial RT	-	-	400
Ellenberg et al. (a) [23]	2009	US/Canada	802	419	353	CNS	Cranial RT	31.5	8.5	277
Ellenberg et al. (b) [23]	2009	US/Canada	5937	2876	3931	Multiple	RT	32.2	-	3489
Evans et al. [24]	1995	UK	48	26	27	Multiple	Multiple	20	-	13
Frederiksen et al. [26]	2022	Scandinavia	10461	5547	9605	Multiple	Multiple	40	-	-
Frange et al. [27]	2009	France	45	26	17	CNS	Cranial RT	25.77	9.07	6
Freycon et al. (a) [28]	2014	France	59	27	41	Heaemato	Stem cell	25.4	8.27	18
Freycon et al. (b) [28]	2014	France	19	-	9	Heaemato	Stem cell	22.47	6.73	-
Frobisher et al. [29]	2008	UK	10257	5256	6462	Multiple	Multiple	32.08	-	-
Gerhardt et al. [30]	2007	US	56	39	44	Multiple	Multiple	18.65	11.36	-
Gray et al. [31]	1992	Canada	62	40	39	Multiple	Multiple	25.59	9.9	21
Green et al. [32]	1990	US	227	122	166	Multiple	Multiple	27.2	11.4	118
Guy et al. [33]	2017	US	239	105	130	Multiple	Multiple	-	-	92
Hayek et al. [34]	2020	US	1041	513	684	Multiple	Multiple	35.54	8.96	523
Hays et al. (a) [35]	1992	US	79	43	68	Multiple	RT	33.9	-	47
Hays et al. (b) [35]	1992	US	111	50	94	Multiple	RT	35.8	-	69
Holmqvist et al. [37]	2010	Sweden	167	81	44	Heaemato	Cranial RT	30	6	8
Howard et al. [38]	2014	US	46	20	20	CNS	Multiple	27	8.5	2
Ishida et al. [39]	2014	Japan	239	123	132	Multiple	RT	24.3	7.5	32
Ishida et al. [40]	2011	Japan	184	76	106	Multiple	Multiple	23.1	8.3	24
Kieffer et al. [41]	2019	France	58	36	23	CNS	Cranial RT	25.1	10.2	-
Kiltie et al. [42]	1997	UK	13	-	2	CNS	Cranial RT	-	-	0
Kim et al. [43]	2013	Korea	223	130	39	Multiple	Multiple	21.92	9.91	6
Kirchhoff et al. [44]	2010	US/Canada	6339	3499	5318	Multiple	Multiple	34.2	-	3053
Kirchhoff et al. [45]	2011	US	5386	2682	4093	Multiple	Multiple	-	-	2860
Kirchhoff et al. [46]	2011	US/Canada	6671	3385	4845	Multiple	Multiple	-	-	3371
Korinthenberg et al. [48]	2011	Germany	29	-	17	CNS	Cranial RT	-	9.17	4
Krull et al. [49]	2013	US	567	270	408	Heaemato	Multiple	33	-	-
Krull et al. [50]	2012	US	62	29	52	Heaemato	RT	42.2	15.1	-
Langeveld et al. [51]	2003	The Netherlands	500	265	265	Multiple	Multiple	24	8	138
Langeveld et al. [52]	2004	The Netherlands	500	265	275	Multiple	Multiple	24	8	135
Lannering et al. [53]	1990	Sweden	23	-	16	CNS	Multiple	24	-	4
Ljungman et al. [54]	2022	Finland	60	39	23	CNS	Multiple	28.1	8.5	16
Löf et al. [55]	2011	Sweden	51	28	24	Heaemato	Stem cell	27	10	20
Lu et al. [56]	2011	US	10397	5593	8279	Multiple	Multiple	-	-	3874
Marina et al. [57]	2013	US	1094	539	932	Bone/Sarcoma	Surgery	32	11	-
Maule et al. [59]	2016	Italy	520	-	117	Multiple	Multiple	-	-	-
Meadows et al. [60]	1989	US	95	50	68	Multiple	Surgery	24.2	6.1	-
Mody et al. [61]	2008	US	1645	888	1554	Heaemato	Multiple	-	-	-
Mostow et al. [62]	1991	US	342	-	290	CNS	Surgery	32	11.3	-
Mulrooney et al. [63]	2008	US/Canada	272	124	244	Heaemato	Multiple	28	7	105
Nagarajan et al. [64]	2003	US	694	353	576	Bone/Sarcoma	Surgery	-	-	-
Nathan et al. [65]	2007	US	1086	451	732	Multiple	Surgery	-	-	445
Nicholson et al. [66]	1992	US	111	97	77	Bone/Sarcoma	Multiple	32.19	14.65	23
Nicklin et al. [67]	2022	UK	69	37	28	CNS	RT	24.6	7.2	-
Nies et al. [68]	2017	The Netherlands	67	9	61	Thyroid	Surgery	35.42	14.61	43
Ottaviani et al. [70]	2014	US	38	-	31	Bone/Sarcoma	Surgery	37.9	13.2	23
Otth et al. [71].	2022	Switzerland	1692	-	1162	CNS	Multiple	24.61	10.57	-
Pang et al. [72]	2008	US	9736	4611	8289	Multiple	Multiple	28.18	10.83	4189
Pastore et al. [73]	2001	Italy	485	208	279	Multiple	Multiple	24.3	7.5	-
Pillon et al. [74]	2013	Italy	141	86	124	Heaemato	Cranial RT	33.35	5.63	45
Pompili et al. [75]	2002	Italy	20	13	10	CNS	Surgery	27.7	8.67	7
Prasad et al. [76]	2015	US	3603	1814	2668	Multiple	Cranial RT	-	-	1325
Pui et al. [77]	2003	US	856	419	510	Heaemato	Multiple	-	5.11	-
Remes et al. [78]	2021	Finland	71	46	28	CNS	RT	27.65	8.4	18
Rourke et al. [79]	2007	US	182	84	121	Multiple	Multiple	22.3	8.7	41
Sato et al. [81]	2018	Japan	78	56	38	CNS	Multiple	23.5	12.7	-
Seitzman et al. [82]	2004	US	578	291	422	Heaemato	Cranial RT	-	-	150
Soejima et al. [83]	2019	Japan	114	-	65	Multiple	Multiple	-	-	-
Strauser et al. [85]	2015	US	385	162	295	Multiple	Multiple	38.39	-	-
Strauser et al. [84]	2019	US	110	52	64	CNS	Multiple	23.05	9.59	-
Stuber et al. [86]	2010	US/Canada	6542	3119	5067	Multiple	RT	31.85	8.21	3322
Sundberg et al. [87]	2010	Sweden	217	105	105	Multiple	Multiple	24	9	71
Sundberg et al. (a) [88]	2013	Sweden	18	10	9	Heaemato	Stem cell	27.33	6.19	6
Sundberg et al. (b) [88]	2013	Sweden	52	25	25	Heaemato	Multiple	24.17	7.5	13
Szilagy et al. [89]	2019	Austria	102	51	94	Multiple	Multiple	32.8	11.2	49
Tebbi et al. [90]	1989	US	40	16	29	Multiple	Multiple	26.2	16.5	22
Tremolada et al. [93]	2016	Italy	205	126	58	Multiple	Multiple	18.96	7.09	-
Tremolada et al. [92]	2015	Italy	213	118	58	Multiple	Multiple	19.4	7.9	-
Van Erp et al. [94]	2022	The Netherlands	151	58	105	Multiple	Surgery	24.1	10.5	75
Wengenroth et al. [95]	2016	Switzerland	1506	787	1174	Multiple	Surgery	29.3	-	-
Winterling et al. [96]	2018	Sweden	59	32	33	Heaemato	Stem cell	28	11	-
Yagci-Kupeli et al. [97]	2013	Turkey	201	126	83	Multiple	Multiple	25.42	9.67	30
Zebrack et al. [98]	2011	US	6425	3064	4999	Multiple	Multiple	32.3	8.7	3731
Zeltzer et al. [99]	2008	US/Canada	7147	3481	5822	Multiple	RT	32.7	6.74	3938
Zeltzer et al. [100]	1997	US	580	293	452	Heaemato	Multiple	22.6	-	185

RT—radiotherapy; CNS—central nervous system; Haemato—haematological cancers. * “Treatment type” should be read as more than 50% of the CCS included received specific treatment. In the case of mixed treatments, unspecified percentages, or those lower than 50%, the term “multiple” has been used.

**Table 2 cancers-14-04586-t002:** Univariate and multivariate random effects meta-regression model.

Factor	*N*. of Studies	β (ES) 95% CI (Univariate)	β (ES) 95% CI (Multivariate) ^a^	*p*-Value (Multivariate)
**Cancer diagnosis**	93			
Multiple		§ (reference)	§ (reference)	
CNS		−0.18 (−0.27/−0.08)	−0.14 (−0.23/−0.04)	0.007
Haematological		−0.04 (−0.13/0.06)	−0.05 (−0.15/−0.51)	0.314
Bone/Sarcomas		0.12 (−0.05/0.3)	−0.02 (−0.24/−0.19)	0.848
**Treatment**	93			
RT		§ (reference)	§ (reference)	
Cranial RT		−0.17 (−0.32/−0.027)	−0.14 (−0.23/−0.076)	0.362
Stem cell transplantation		−0.15 (−0.33/−0.04)	0.01 (−0.17/0.19)	0.895
Surgery		0.06 (−0.08/0.21)	0.12 (−0.04/0.25)	0.151
Multiple		−0.04 (−0.15/0.07)	0.05 (−0.05/0.14)	0.309
**Diagnosis and treatment**	75			
Before 1990		§ (reference)	§ (reference)	
After 1990		−0.19 (−0.28/−0.10)	-	-
**Geographical area**	93			
North America		§ (reference)	§ (reference)	
Europe		−0.13 (−0.21/−0.06)	−0.07 (−0.14/−0.03)	0.063
Asia		−0.24 (−0.38/−0.09)	−0.14 (−0.28/−0.002)	0.047
**Mean age at investigation**	76	0.02 (0.014/−0.027)	0.02 (0.01–0.023)	0.00
**Male %**	85	−0.01 (−0.00/−0.01)	−0.01 (−0.005/0.001)	0.15
**Married %**	59	0.01 (−0.005/−0.01)	-	-
**Graduated %**	61	0.01 (0.00/0.0003)	-	-

Abbreviations: RT—Radiotherapy; CI—Confidence Interval. ^a^ adjusted for all other variables except for marital status and educational level.

## Data Availability

The data presented in this study are available on request from the corresponding author.

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
