# Peer review of "Employment among Childhood Cancer Survivors: A Systematic Review and Meta-Analysis"

_cancers, 2022, doi:10.3390/cancers14194586_

Round 1

Reviewer 1 Report

I enjoyed reading this paper and it is refreshing to be able to say that I have virtually nothing on which to comment. The subject is important and scientific work is meticulous. I congratulate the authors on their excellent paper.

My only comments are:

1. What happened to the students? In line 103 it is said that the number of students was recorded. Unless I have missed something, they are not mentioned thereafter. Although the definition of work in the analysis excludes them, if they were counted it would be interesting to know how many there were. What %age were they in the various cohorts? The analysis shows that more education leads to more employment.  If the figures are easily available, I would recommend their inclusion. [They may be in the supplementary material which I could not find].

2. I had difficulty understanding lines 277-280. Does it mean that in reference 106 noncranial DXT and chemo did affect employment while the rest of the data analysed by the authors did not? Clarification please.

Author Response

Dear reviewer, thank you for your interest in reviewing our manuscript. We have appreciated all your comments, which have been considered carefully and answered in the attached file.

We think and hope this revision has improved the quality of our manuscript

Regards

Reviewer 2 Report

Thank you for the opportunity to review this manuscript. While the subject is interesting, I have several concerns regarding the scientific quality. Firstly, the study does not really address a knowledge gap in the literature. The authors write that this, to their knowledge, is the first study investigating employment rates in survivors of childhood cancer. However, several systematic reviews have been published on this subject (see for example Mader et al, 2017). Secondly, the study has several methodological problems. The most pressing ones are:

1.       The reference to the quality assessment [11] is incorrect and the reference does not lead you to the quality of assessment table they are using. And the quality assessment method used (Joanna Briggs, or Munn 2014) is remarkably different from the one the authors have used. One of the most noteworthy alterations is that the “not applicable” column is entirely missing. Why did the authors remove this? There is a vast difference between, for example, not conducting sample size calculations because it’s not relevant for the research question and because of a substandard method. Why have these been scored the same way?

2.       Relatedly, there is no reference for the grouping of scores. What are these based on? In the quality assessment template that they have been using, there is nothing that explains why, for example, a score of 8 should be considered “good”. Is this arbitrary, or was it based on previous research? If the latter, please provide a reference.

3.       The authors do not disclose how inclusion of the studies were conducted. Was one researcher’s opinion necessary? If more than 1 was needed, how did you resolve conflicts? This type of information is essential for a systematic review and one of the basic criteria.

4.       The Prospero registration has not been correctly updated. According to the latest update (July 18th), data extraction for this study has not begun. Note the following text in Prospero:
“The record owner confirms that the information they have supplied for this submission is accurate and complete and they understand that deliberate provision of inaccurate information or omission of data may be construed as scientific misconduct.”

Please update your Prospero registration so that it is align with your progress, as it might otherwise come across as scientific misconduct.

5.       Please consider using a different word than “prevalence” for employment rates. In medicine and epidemiology, the word prevalence is reserved for medical conditions and diseases. It is therefore inappropriate to apply it to a socio-economic variable such as employment. Furthermore, it was not one of the search terms, which makes it appear even more unrelated to your subject. The title simple doesn’t correspond to the content of the manuscript or the methodology. May I suggest using “frequency” or “rate”?

Author Response

(The authors gave the same response as above.)

Reviewer 3 Report

Thank you for the opportunity to review this manuscript. Overall, the topic is interesting and compelling. However, there are several concerns I have about the presentation of this work, which I will detail below.

1) Methods are not clear in regards to what exactly "counts as work". Without clarity in this, interpretation of results and discussion is severely limited.

2) Table 1 needs to be improved. It is not comprehensible as currently presented. What do the numbers represent? Footnotes and additional information is needed.

3) I appreciate the challenge of identifying enough studies, but I would be interested in the inclusion of culture shift considerations that make the findings comparable across 30+ years.

4) Several statements throughout the discussion are not supported and appear out of the blue. I.e.,  "CCS may experience such identity problems." Lines 352-353. What does this statement mean?

5) Is this the first review of its kind or were there others as well? (Line 259).

6) The utility of the quality analysis that was conducted is unclear. How different would the results be without it?

Author Response

(The authors gave the same response as above.)

Round 2

Reviewer 2 Report

Thank you for the opportunity to review this revised version of the manuscript. The revisions have improved the quality of the manuscript, but I still have concerns about the quality assessment table and, relatedly, the lack of information and discussion regarding that table. Even though the authors didn’t consider N/A to be applicable for any of the included studies, this must still be disclosed in the paper (preferable both under the table and in the text. Secondly, it is still not clear in the paper why the manuscripts were given a summary score. In what ways this this add to the study? Since a summary score IS used, it should be justified in the papers for the readers, and not only in the response to reviewers. Additionally, since the summary score is used, the results should be addressed in the discussion section. For example: 1) The average summary score was fair, but what did the 6 included studies with poor quality have in common? 2) Do you have any recommendations for future studies based on the outcome of the quality assessment? 3) What does it say about the field of research that questions 4 (study participants), 7 (reliable measurement), and 8 (statistical analysis) so often were scored as 0?

Author Response

(The authors gave the same response as above.)

Reviewer 3 Report

Thank you for your prompt attention to each of the concerns and comments I made during my initial review of your manuscript. I am satisfied with your responses and pleased with many of the edits you made. 

The only, remaining, ever-so-minor edit I have is in line 57, remove the first use of "reference" from the sentence and leave in the word "period", so it reads: "...during a short reference period..." Or something comparable to make the sentence more readable.

Author Response

(The authors gave the same response as above.)

Round 3

Reviewer 2 Report

I have no further comments. I am satisfied with the revisions of the manuscript.